# Role of Long Chain Fatty Acids in Developmental Programming in Ruminants

**DOI:** 10.3390/ani11030762

**Published:** 2021-03-10

**Authors:** José Alejandro Roque-Jiménez, Milca Rosa-Velázquez, Juan Manuel Pinos-Rodríguez, Jorge Genaro Vicente-Martínez, Guillermo Mendoza-Cervantes, Argel Flores-Primo, Héctor Aarón Lee-Rangel, Alejandro E. Relling

**Affiliations:** 1Facultad de Agronomía y Veterinaria, Universidad Autónoma de San Luis Potosí, San Luis Potosí 78321, Mexico; alejandro.roque@uaslp.mx (J.A.R.-J.); hector.lee@uaslp.mx (H.A.L.-R.); 2Facultad de Medicina Veterinaria y Zootecnia, Universidad Veracruzana, Veracruz 91710, Mexico; zS18015915@estudiantes.uv.mx (M.R.-V.); jpinos@uv.mx (J.M.P.-R.); jvicente@uv.mx (J.G.V.-M.); argflores@uv.mx (A.F.-P.); 3Centro de Investigación en Micología Aplicada, Universidad Veracruzana, Xalapa, Veracruz 91010, Mexico; guimendoza@uv.mx; 4Department of Animal Sciences, The Ohio State University, Wooster, OH 44691, USA

**Keywords:** fetal programming, omega-3, omega-6, ruminants

## Abstract

**Simple Summary:**

The objective of the current review is to provide a broad perspective on developmental program aspects of dietary n-3 FA supplementation in ruminants during pre-conception, conception, pregnancy, early life, including its effects on production, lipid metabolism, and health of the offspring. Offspring growth and metabolism could change depending on the FA profile and the stage of gestation when the dam is supplemented. Despite this extended review we are highlighting areas that we consider that there is a lack of information.

**Abstract:**

Nutrition plays a critical role in developmental programs. These effects can be during gametogenesis, gestation, or early life. Omega-3 polyunsaturated fatty acids (PUFA) are essential for normal physiological functioning and for the health of humans and all domestic species. Recent studies have demonstrated the importance of n-3 PUFA in ruminant diets during gestation and its effects on pre-and postnatal offspring growth and health indices. In addition, different types of fatty acids have different metabolic functions, which affects the developmental program differently depending on when they are supplemented. This review provides a broad perspective of the effect of fatty acid supplementation on the developmental program in ruminants, highlighting the areas of a developmental program that are better known and the areas that more research may be needed.

## 1. Introduction

The maternal conditions and nutrition during pregnancy are critical for fetal development. Adverse effects on maternal physiology during gestation are associated with poor health of the offspring in adult life [1]. The term “fetal programming” (FP) was defined in the early 1990s as a phenomenon linking long-term adverse health consequences in animal species with adverse nutritional exposures in utero [2]. This FP theory was first proposed by David Barker; and FP and its long-term impacts have been evaluated particularly from a human health and disease perspective [3]. Such studies have revealed that FP has implications for a wide range of body functions, which are also key determinants of animal growth and productivity [4]. However, knowledge about the potential long-term implications of FP for animal productivity is still limited. Such knowledge is needed in order to assign the best management strategies to minimize implications of adverse FP for animal productivity and avoid the possible trans-generational transfer of undesirable FP outcomes [2]. Recent research has demonstrated that maternal circulating concentrations of essential fatty acids (FA) can modulate fetal growth and development [1].

Fatty acids, omega-3 (n-3), and omega-6 (n-6) are essential for normal physiological functioning and for the health of non-ruminants and ruminants. Fatty acids have multiple functions in the body, but, in this review, we will focus on the developmental programming effect of FA in ruminants, considering the effect of FA supplementation on imprinting during gametogenesis, embryo, fetal, and postnatal development. Current research has demonstrated beneficial effects of polyunsaturated FA supplementation, especially n-3 FA, during pregnancy in ruminants [5,6,7,8,9,10,11,12].

The objective of the current review is to provide a broad perspective on developmental program aspects of dietary FA supplementation in ruminants during pre-conception, conception, pregnancy, early life, including its effects on production, metabolism, and health of the offspring.

### 1.1. The History of the Importance of Fatty Acid in the Diet

Fatty acids play important roles in fetal growth, gene expression of genes involved in lipid metabolism, cognitive and behavioral development, and energy metabolism in mammals [13]. The maternal condition during pregnancy is critical for fetal development, with adverse effects on maternal physiology associated with poor health of the offspring during adult life [1]. Imbalance in fatty acid intake during fetal development may cause structural and functional changes in metabolism [14]; these changes guarantee the survival of the embryo or fetus, but also has long-term effects on the adult life of the animal.

A requirement for specific fatty acids in the diet was first described for growing rats by Burr and Burr [15]. It was shown that skin lesions and physiological symptoms of deficiency, including kidney damage and death, could be reversed by supplementing a diet with lard [15]. In further studies [16,17], the researchers found that linoleic (LA) and linolenic (LNA) acids contributed to decreased deficiency symptoms. There was a great progress in the 1960s on research in essential fatty acids (EFA). Holman [18,19] feeding pure LA, LNA, and other fatty acids to rats showed that dienoic LA can be converted to tetraenoic arachidonic acid (AA), trienoic LNA can be converted to penta- and hexanoic acids, and products of the two precursors cannot be interconverted. Because traditional chemical terminology at that time causes confusion when considering the interconversions of the EFA, Holman [20] introduced and proposed the concept of grouping EFA by families according to the position of the terminal double bond in the molecule (“omega families”). This terminology continues in the actual literature, whereas the proper chemical designation for these families is “n-“ or omega. A concise summary of the evolution of concepts in EFA metabolism and the slow acceptance of the role of n-3 FA in mammals health was presented by Holman [20]. Although investigators recognized the role of EFA to decrease biological issues in mammals, the lack of knowledge of their metabolism has a limited understanding of the physiological basis for their essentiality [13].

The increased intake of n-3 FA, especially eicosapentaenoic acid (EPA; C20:5 n-3) and docosahexaenoic acid (DHA; C22:6 n-3), is associated with the development of retinal and brain membrane phospholipids; and it is involved in visual and neural function and neurotransmitter metabolism [21], as well as increased cognitive behavior and lipid metabolism during early life [22] in mammals. Additionally, FA intake during pregnancy has been reported as an essential nutrient [23]. The placenta is the key organ through which nutrients such as these FA flow from the mother to the fetus [4,23]. The maternal FA passes through the placenta and storage in different tissues in the fetus [23]. In addition, the FA intake could modify the FA profile in the colostrum and milk [24]. In prenatal stages, fatty acids and metabolites play an important role in cell enlargement, differentiation, and regulatory response between metabolic and neuroendocrine environments [25]. Fatty acid contributes to the molecular signals that regulate appetite and energy metabolism. In addition, fatty acids act as metabolic sensors that participate in the regulation of genes involved in energy oxidation and storage [4]. Therefore, it is thought that insufficient, imbalanced, or excessive fatty acid intake in the early stages of development can also contribute to metabolic (or nutritional) programming [24].

### 1.2. Fatty Acids in Ruminant Nutrition

Based on their chemical structure, FA can be differentiated into three main groups: (1) saturated fatty acids (SFA), no double bonds; (2) monounsaturated FA (MUFA), only one double bond present; and (3) Polyunsaturated FA (PUFA), at least two double bonds present. In addition, long-chain FA can be divided into three main families, depending upon the site of the first double bond from the methyl side: n-3, n-6, and n-9 [13]. All n-3 and n-6 FA are PUFA, whereas most of the n-9 FA are MUFA (such as oleic acid, C18:1n-9) [26]. Both n-3 and n-6 FA are considered essential FA because they cannot be synthesized de novo by mammals. The initial 18-carbon molecule of each class can undergo de novo elongation and further desaturation [13]; however, the position of the first double bond remains the same.

Several studies have been conducted to analyze the biological activities of FA during pregnancy in non-ruminants and ruminant models [13]. Nevertheless, an important difference between non-ruminants and ruminant models has been the type of digestive system and the metabolism of the lipids [27]. Ruminants possess a forestomach anaerobic fermentation system (rumen-reticulum), anterior to the gastric organ and intestines. Dietary triglycerides are entirely hydrolyzed in the rumen, then unsaturated FA are hydrogenated; afterward, FA adsorbed on feed particles are absorbed in the small intestine [28]. Ruminants diets typically have a low fatty acid concentration (2.5–3.5% of dietary dry matter intake; DMI) [26]. The most common sources of lipids in ruminants feed are oilseeds, several plants, calcium salts, and animal by-products that contain fat or oil, such as fish meal and distillers’ grains. On developmental studies, there are a few reports that prove the source of FA and the effects on developmental programming [27]; and they are not studies that compared FA sources. Santos et al. [29] concluded that the absorbed FA profile is more important than the FA source to modulate physiology in dairy cattle. However, there are no studies reported associating the specific fatty acid absorbed and the developmental program effect on the offspring life, or the association of rumen environment, fatty acid metabolism, and the developmental program.

## 2. Fatty Acid Supplementation before and during the Conception in the Developmental Program

Fatty acids are known to regulate metabolism [13]. Imbalance in FA intake during development causes structural and functional changes in metabolism [13]. The fetus adapts to maternal FA intake through changes in the production of fetal and placental hormones that regulate metabolism, redistribute blood flow, and control growth [4,7,8]. However, before fetal programming, the supplementation with FA has been linked with fertility, FA profile, and DNA modifications in sperm and ovules that transmit acquired phenotypes—more recently, RNA modifications [30].

### 2.1. Supplementation with FA to Male Ruminants

An experiment on paternal effects on developmental programming (where the male is exposed to a specific environment before mating) provided evidence that sperm-borne factors responsive to changes in dietary FA can modulate the developmental programming of the offspring. These changes are called epigenetic inheritance, a term referring to the direct modification of the gametic epigenome by the semen quality with specific epigenetic marks and subsequent transmission to the next generation [31]. A high rate of male fertility is desirable in the ruminant industry to get high quality semen for successful artificial insemination or natural mating [32]. Semen can play a vital role to improve farm economic profit through epigenetic changes that can be inherited in the offspring [33]. However, semen is influenced by many factors such as genetics, management, environment, and nutrition [32]. Fatty acids serve as a source of energy and are critical components of the physical and functional structure of cells [13]. Three major points have been pivotal in investigating the field of paternal effects and the relation with FA profile: (1) the comparison between FA profiles on fertility [34], (2) the effect of dietary FA on sperm FA profiles as well as sperm quality and quantity [32], and (3) dietary FA as a promoter of establishment of epigenetic marks by stimulating expression of specific genes during offspring development [30]. All of the points previously described have specific relevance in the ruminant industry. For the first and second points, previous to the mating, the role of the FA in the sperm quality has been the most described [34]. Little information is available with regard to the effect of dietary FA and epigenetic changes in offspring using ruminants such as animal models [35]. Nevertheless, for mammals, a few pieces of evidence have described the role of PUFA in the spermatozoa epigenome, in particular, DNA oxidation, DNA methylation, and gene expression associated with lipid metabolism [30]. The fragmentation of sperm DNA affects post-implantation embryonic development; specifically, high levels of sperm DNA fragmentation can compromise the viability of an embryo, resulting in pregnancy loss [36]. Sperm carrying damaged DNA can complete the initial process of fertilization; however, the developmentally necessary genes in the damaged sperm DNA may hinder embryonic development upon activation of the embryonic genome [37]. Furthermore, sperm DNA damage has been linked to delayed chromosomal instability in blastocysts and post-implantation developmental abnormalities [37]. However, very little is currently known regarding the nature of germline DNA damage and the extent to which germ cells are capable of eliminating damaged DNA and completing the process of DNA repair [38]. Despite the reports that describe in animal models that paternal inheritance can provide epigenetic inheritance [39], there is no evidence that supplementation with FA on males can affect the developmental program and offspring development.

Methylation of DNA is a key component of the full array of epigenetic phenomena that works together to establish and maintain a gene expression states [40]. It has been described that the ruminants DNA is methylated at the 5-position of cytosine residues predominantly within guanine-cytosine (CpG) dinucleotides such that about 60–80% of CpGs within the genome are methylated [41]. This definition of epigenetics does not apply to spermatozoa, which are transcriptionally inactive and represent the ultimate form of cell differentiation, destined to give rise to a new individual after fertilization of an oocyte and not to daughter cells with a similar phenotype [42]. The differentiation of male germ cells into functional spermatozoa requires a unique epigenetic reprogramming involving large-scale DNA methylation changes, the replacement of most histones by protamines, and an accumulation of specific non-coding RNAs [32]. Even though transcription is barely detectable in mature sperm cells, the male germline differentiation programmed is orchestrated by a dynamic sequence of transcriptional regulations that are directly reliant on epigenetic reprogramming [42]. The biological processes described previously has been recognized with evidence supporting the biological epigenetic process [41]; however, a lack of evidence of the role of FA supplementation and its effects on the genome-wide DNA methylation profiles are still scarce and limited [30]. Further research needs to describe the epigenetic of genome-wide DNA methylation through epigenetic biomarkers, which can help to define an amount and a time of when FA supplementation might be required to improve the offspring’s performance.

### 2.2. Supplementation with FA on the Flushing Period in Female Ruminants

Among their physiological roles, n-3 PUFA affects reproduction in female ruminants. A review by Moallem [26] provides an extensively broad perspective and demonstrates the importance and potential of n-3 FA dietary supplementation on reproduction during the flushing period in female ruminants. Moreover, a few pieces of data have been published describing the importance of FA, type, and amount in the oocytes and as a modulator of developmental programming. Most of the information has been described in humans [43,44] and some in Holstein cows [45,46]. In order to present a viable gamete for fertilization, O’Doherty et al. [45] described that the oocytes must complete a series of critical processes during the transition from the resting primordial stage to the preovulatory stage. These changes include proliferation and differentiation of cytoplasmic organelles, synthesis, and storage of mRNA and proteins required to drive the initial cell cycles of embryogenesis, resumption of/and completion of meiosis, and attainment of epigenetic modifications [45]. One such epigenetic process is the essential reprogramming of genomic imprints, which occurs during the mammalian oocyte growth stage [47]. This process involves reprogramming the maternal genome, through DNA methylation of differentially methylated regions (DMRs), to elicit parent-of-origin specific expression of a small group of genes, collectively known as imprinted genes [40]. Inappropriate methylation at imprinted loci can lead to ectopic expression of imprinted genes (both copies silenced or expressed) and has been observed in a number of developmental and pathological conditions, such as Beckwith–Wiedemann syndrome, Prader–Willi syndrome, and Angelman syndrome, spontaneous abortion, early embryonic lethality, and transient neonatal diabetes [45]. Recent studies have shown that an adverse metabolic environment induced by lactation alters the metabolomic, steroidogenic, and transcriptomic profile of ovarian follicles during their development in postpartum lactating cows compared to non-lactating heifers [46,48]. These differences are characterized by higher concentrations of SFA, such as palmitic and stearic acids, lesser concentrations of n−3 PUFA, increased concentration of glycine and L-glutamine, and a small amount of L-alanine and oxoproline in follicular fluid from lactating cows compared to those from dry heifers [48]. For comparison with these results and to describe a possible genomic imprinting, O´Doherty et al. [45] collected oocytes and follicular fluid samples from cows between days 20 and 115 post-calving. In a complimentary study, cumulus oocyte complexes were in vitro matured under high non-esterified fatty acid (NEFA) concentrations and in the presence of the methyl-donor S-adenosylmethionine (SAM). The authors observed a loss of DNA methylation in the PLAGL1 locus in oocytes, following in vitro maturation (IVM) in the presence of elevated NEFAs and SAM. Metabolomic analysis of postpartum follicular fluid samples revealed significant differences in several branched chain amino acids, with fatty acid profiles bearing similarities to those characteristics of lactating dairy cows. Finally, they concluded that the postpartum ovarian environment may affect maternal imprint acquisition; and elevated NEFAs during IVM can lead to the loss of imprinted gene methylation in bovine oocytes. This finding suggests that FA are important in the regulation of pre-mating developmental programing; as for semen, there is no information on the FA profile, time, and amount of supplementation on the different ruminant models (beef, dairy, or sheep).

### 2.3. Supplementation with FA on the Conception Period

Mammalian development is characterized by bimodal DNA methylation reprogramming that occurs initially during germ cell development and then during preimplantation [49,50]. Primordial germ cells (PGCs) enter the developing germinal ridge and begin differentiation and expansion. At this time, the highly methylated PGCs undergo rapid genome-wide demethylation such that, by day 12.5, most of the methylation is lost [50]. This reprogramming phase coincides with the erasure and resetting of parent-of-origin specific marks that include DNA methylation of imprinted DMRs associated with allele-specific gene expression [49,51]. The exact timing of de novo methylation has not been firmly established but is initiated in males at 14.5 days post-mating and thereafter in females such that the mature gametes of both sexes will eventually become highly methylated. The second phase of reprogramming methylation occurs between fertilization and formation of the blastocyst. In fertilization, a rapid paternal-specific asymmetric loss of methylation is observed [49]. This process takes place in the absence of transcription or DNA replication and is called active demethylation. Thereafter, there is a stepwise decline in methylation until the morula stage [49,50,51]. This decline occurs as a result of the absence of the primary DNA methyltransferase, DNMT1, during DNA replication [50]. These biological processes represent a unique opportunity for epigenetic modifications through the maternal diet [52]. Differences in the effect of FA composition in feed ingredients under diverse environmental conditions have not been investigated in depth [53]. Further research is needed to describe the effect of FA composition in different geological locations on the development program at the conception period.

## 3. Effect of FA Supplementation during Gestation on the Developmental Program

### 3.1. Placental FA Transport

In humans, EFA, triglycerides (TG), and lipoproteins consumed during pregnancy reach the fetus by passing through the placenta via special transports [4]. In particular, fetal accretion of DHA is essential for early development on the fetal central nervous system [54]. The placenta normally mediates adequate delivery of physiologically important PUFA to the fetus by extracting and transporting fatty acids in a directional, preferential, and timely manner [55]. Epitheliochorial placentas are less permeable to free FA than hemochorial ones [56], and the placental transport of short-chain and long-chain FA in ruminants might be limited [57]. Campbell et al. [58] described that FA in maternal circulation is the main source of FA for the fetus. The role of n-3 FA in the unborn fetal ruminant’s development and the limited transport of long-chain FA through the ruminant’s placenta raise an interesting question with regard to supplementation of these unique FA to unborn ruminants. Jones et al. [54] described the interaction between FA, placenta uptake of FA, and the function of placental lipoprotein lipase (pLPL) and epithelial lipase. Both enzymes are present in the maternal-facing microvillous membrane of the syncytiotrophoblast; and they release fatty acids from maternal circulating TG rich lipoproteins to allow placental uptake of NEFA [59]. Concentrations of NEFA, DHA, and AA are three- to four-fold greater in the placental intervillous space than in the maternal circulation, suggesting placental lipases selectively release long-chain PUFA from TGs. Indeed, pLPL is known to preferentially hydrolyze fatty acids in the second position of TG, which tends to be less saturated [60]. Duttaroy et al. [61] reported that placental lipase activity increases during the final trimester of pregnancy, probably serving to enhance placental fatty acid delivery during the period of the maximal fetal fatty acid requirement.

Omega-3 FA may enter the placental syncytium by passive diffusion or via several membrane-bound carrier proteins (Figure 1) including fatty acid translocase (FAT/CD36), fatty acid transport proteins (FATP1–6), plasma membrane fatty acid-binding protein (FABPpm), or placental plasma membrane FABP (p-FABPpm) [62]. Although the specific mechanisms of these proteins in placental fatty acid uptake, metabolism, and transfer are not fully known, directional and preferential transfer of fatty acids across the placenta can be attributable to differences in the affinity of FA for these proteins. Once within the cell, FA bind to cytosolic FABP which facilitate intracellular fatty acid movement and interactions with subcellular organelles [59].

The FABP have an increased affinity for FA with increasing chain length; however, binding affinity is tissue-specific [54]. Jones et al. [54] described that p-FABPpm exhibits higher affinity and binding capacity for DHA and AA compared with OA and LA. This carrier protein is also exclusively found in the microvillous membrane of the syncytiotrophoblast (Figure 1), potentially driving the unidirectional transfer of long chain polyunsaturated fatty acids (LC-PUFA) from maternal to fetal compartments. Moreover, Desantadina et al. [63] described the effect of time of pregnancy on fatty acid transporter mRNA expression in maternal (cotyledons) and fetal bovine placenta (caruncles); the genes evaluated were FATP-1, FATP-4, and FABP-1. They reported that FATP-1 mRNA expression was greater in cotyledons on the first third of pregnancy as compared with the concentration in caruncles. In addition, the difference in FATP-1 mRNA expression in cotyledons decreased, reaching a similar concentration to that observed in caruncles. Finally, they conclude that FATP-1 might play an important role in fatty acid transport during early fetal development.

Additionally, in a study conducted with ewes using Ca salts enriched with EPA and DHA or SFA and MUFA, the treatment enriched with SFA and MUFA showed that free FA receptors are increased by supplementation during early gestation in maternal (cotyledons) and fetal (caruncles) compartments, and could be regulating lipid metabolism and placenta functions [9] (Figure 1).

Something to consider is that the FA profile on the newborn depends on the FA passage through the placenta and the fetus FA metabolism. Moallem and Zachut [64] added one of the three supplements to the diet of late pregnant cows: encapsulated SFA, flaxseed oil, or fish oil on day 256 of pregnancy. Blood samples were taken from the newborn calves immediately after delivery and before offering colostrum. The proportion of DHA was 1.9 times greater in calves born to cows supplemented with fish oil during late gestation than the ones born to cows fed the other two supplements, but no differences were observed in the plasma proportions of ALA, EPA, or docosapentaenoic acid (DPA) between animals. A similar effect of the maternal diet on the FA profile in newborn plasma was also observed by García et al. [65]. In a recent study [66] in beef cattle, where dams were supplemented with either a source of PUFA (rich in linoleic acid, eicosapentaenoic acid, and docosahexaenoic acid) or SFA/MUFA (rich in palmitic and oleic acids), PUFA steers had greater plasma EPA and lower plasma linolenic acid and oleic acid than SFA/MUFA steers. As described previously, maternal-newborn FA transfer could be then considered to be dependent on the concentration of gradients across the placenta. The wide difference between the FA profiles of mothers and calves observed by Moallem and Zachut [64] could be explained by the placental desaturation of FA or by the selective absorption of FA by FA-binding proteins. In humans, it has been demonstrated that placental transfer of DHA involves a multiphase process of intercellular absorption and translocation by various systolic and FA-binding proteins membrane-associated that prefer n-6 and n-3 FA to non-essential FA [24,62]. Likewise, plasma membrane FA-binding protein has been identified in the sheep placenta. The binding of oleate (14C) to placental membranes was reported to be time- and temperature-dependent in sheep [67] without yet identifying one of these proteins in the bovine placenta [26].

### 3.2. Nutritional Programming Effect of FA Supplemented in Early Gestation

Omega-3 FA have beneficial effects on the support of normal growth and development of various tissues during fetal development in the first third of gestation [4]. Some of the effects of DHA and EPA are important for fetal brain development, specific components of the neural system, retinal maturation, and neonatal behavior [24,57]. In recent research conducted in our laboratory, Roque-Jiménez [9] and Oviedo-Ojeda [12] supplemented ewes during the first third of gestation with 1.5% Ca salts of FA as a source of MUFA or PUFA with the objectives of evaluating the concentration of EPA and DHA in the fetal liver (FL) and fetal central nervous system (FCNS), relative mRNA amount of genes associated with transport and metabolism of FA in the FL and placenta, and finally, evaluate the productive performance and hypothalamic neuropeptides on the offspring. Different results were observed. The concentrations of C18:1 isomers increase in the FL and FCNS with MUFA supplementation; however, the FL and FCNS had a greater concentration of C20:3(n-6), C20:3(n-3), C22:1, C22:5, and C22:6 with PUFA supplementation. In the FL, the relative abundance of LPL mRNA was greater due to SFA and MUFA supplementation [9]. As previously described in the study by Desantadina et al. [63], placenta samples were collected in two sections, the cotyledon and caruncles. The results showed that, in the placenta, there was an FA x tissue interaction for relative abundance of DNMT3b in the caruncle of fetuses from SFA and MUFA supplemented dams and the relative abundance of FFAR-4 mRNA transcripts was greater in the caruncle; and for fetuses from PUFA-supplemented dams, the relative abundance of FFAR-4 mRNA transcript was greater in the cotyledon [9]. Fetuses from SFA and MUFA-supplemented dams had a greater relative abundance of FABP-4 mRNA. Results indicate supplementation with PUFA during early gestation increases the total EPA and DHA in the FL. For the placenta, PUFA supplementation led to an increase in the relative abundance of lipid mRNA for transport genes. At this time, we concluded that FA concentration in the liver and brain was not associated with changes in expression of some of the liver and placenta genes, but there might be a mechanism by which supplementation of SFA and MUFA may have increased the expression of lipid transport genes. Later, during the offspring evaluation [12], there was a dam supplementation (DS) × lamb supplementation (LS) interaction for body weight, where lamb supplementation with PUFA born from DS with SFA and MUFA were heavier than the other treatments (Table 1). Lambs born from DS with SFA and MUFA have greater dry matter intake than the offspring born from DS with PUFA. Lambs born from SFA and MUFA supplemented dams had a greater hypothalamus mRNA expression for *cocaine and amphetamine regulated transcript (CART)*, *growth hormone receptor (GHR)*, *metastasis suppressor 1 (KISS1)*, *leptin receptor (Lep-R)*, *pro-opiomelanocortin (POMC)*, and *Neuropeptide Y (NPY)* (Table 1). These results may indicate that supplementation with FA in early gestation plays an important role in developmental programming, also that supplementation with FA during early gestation changes productive performance and neuropeptides’ mRNA expression of lambs independently of the finishing diet. However, the mechanisms that regulate these changes are still unknown, and more studies should be done to understand the mechanism regulating the increase in lamb performance due to FA supplementation in early gestation.

### 3.3. Nutritional Programming Effect of FA Supplemented in Late Gestation

Nutritional management during the last third of gestation has impacted offspring performance, and physiology responses through fetal programming effects (Table 1). More specifically, supplementation with a source of FA during late gestation might have fetal programming effects. The last third of gestation is when the increase of LC-PUFA in the fetal circulation occurs; thus, an increase in maternal dietary supply during late gestation might have a direct effect in intrauterine growth and fetal development affecting offspring subsequent development and welfare [68]. Previous research has demonstrated the beneficial effects of supplementing a source of LC-PUFA during the last third of gestation in fetal development, energy metabolism, mRNA expression, global DNA methylation, and inflammatory response, as well as in offspring growth [69], energy metabolism [8,10,11,65,70], carcass characteristics [70,71], and in the long-term effects in the performance of the offspring [69] (Table 1). Therefore, in utero changes with subsequent possible lasting effects in metabolism and performance of the offspring, by altering maternal diet in late gestation, can be a management practice to enhance animal health and productivity of food producing animals.

Regarding the effects of the addition of PUFA in the diet of the dam during the last third of gestation on the development of the progeny, an increase in growth has been reported in the postnatal period in different ruminant models [8,69,70]. Rosa-Velázquez et al. [10] did not report differences in fetal weight when ewes were supplemented with a source of n-3 PUFA during late gestation. In sheep, lambs born to ewes fed n-3 PUFA during the last third of gestation showed an increase in body weight at birth [7] and during the finishing period [7,8,10] (Table 1). Supplementation during late gestation and early lactation with a source of n-3 PUFA has also affected lamb growth during the preweaning period [72,73] (Table 1). Similarly, Santos et al. [69] reported an improvement in growth performance of calves born from dairy cows fed PUFA at the end of gestation, observing a greater body weight and a better body condition at calving. In another study conducted in dairy cattle, supplementing multiparous cows with fats differing in FA profile, either SFA or PUFA, but providing similar amounts of energy, also increased birth weight compared with calves born to dams not supplemented with fat [65]. The calves born from dams supplemented with SFA also had greater feed intake and average daily gain (ADG) through 60 d of age [74] (Table 1). These findings are consistent with that reported by other researchers in studies conducted in beef cattle who observed an increase in body weight at birth [75], and body weight and average daily gain during the end of the finishing period [70] in calves born to beef cows that were supplemented a source of FA during late gestation (sunflower seeds, and PUFA, respectively). A recent study conducted in beef cattle, Brandão et al. [76] reported no treatment differences in offspring birth weaning body weight, and final preconditioning body weight when cows were supplemented either Ca salts of soybean oil (source of n-6 PUFA) or prilled saturated fat in late gestation (Table 1). Contradictorily, another study conducted by Ricks et al. [77] reported greater preweaning growth in calves from cows fed the same source of n-6 PUFA during gestation (Table 1). The increase in preweaning weight observed by Ricks et al. [77] was mostly noted in primiparous cows and associated with improved milk production from supplementing fat to growing females, whereas Brandão et al. [78] used multiparous cows only. However, most of the results reported in the pre-weaning period on growth performance indicate that supplementing late-gestating beef cows or ewes with a source of PUFA did not impact offspring birth body weight as well as growth from birth to weaning when compared with no-supplemented cohorts [5,8,70,79,80]. Different from previous studies, Jolazadeh et al. [81] observed an improvement in both weaning and final body weight, overall average daily gain, hip height, and wither height of calves born to cows supplemented with lipids (either n-3 or n-6) during late gestation compared to calves born to non-supplemented dams. Jolazadeh et al. [81] reported that the observed changes occurred without changing calves feed intake, but with a tendency for a better feed efficiency for pre-weaned calves born from dams fed fat supplements compared with those born from non-supplemented dams. Regarding the average daily gain, calves born from fat supplemented cows were heavier than those born from non-supplemented dams during the pre-weaning period, but not during the post-weaning period. However, Jolazadeh et al. [81] noted that the benefit of prepartum fat supplementation reported during the preweaning period had a direct impact on the overall benefit of diets in average daily gain for the whole experimental period.

Interestingly, a treatment by sex interaction was reported in two recent studies conducted in two ruminant models [76], where an increase in body weight during the finishing period was reported after maternal FA supplementation during late gestation. In beef cattle [76], a treatment by sex interaction for feed yard ADG, and final BW was reported, these responses were greater in steers from n-6 cows compared with those born to SFA supplemented cows and did not differ between heifers. Brandão et al. [76] noted that the reason for the treatment by sex interaction on average daily gain reported in their study was unclear, since steers and heifers were equally managed as a single group in the feed yard. In sheep, a study conducted in our lab [10] also reported a dam treatment by offspring sex. Ewe lambs born to MUFA supplemented dams were heavier than wethers born to MUFA dams, while wethers born to n-3 PUFA supplemented dams were heavier than ewe lambs born to n-3 PUFA supplemented dams at 60 days of age. In summary, wethers born from PUFA supplemented dams had the greatest post-weaning BW during the finishing period. These results suggest that FA supplementation could affect offspring performance in a sex-dependent manner. The data in Rosa-Velazquez [10] study also suggest that maternal supplementation with SFA + MUFA during late gestation could have a limiting effect on male offspring development, and similarly for female offspring whose dam is supplemented with a PUFA gestational diet. However, there is no data that could help us to explain the physiological mechanism for the interaction of maternal diet and sex effects. One of the most interesting effects of altering the growth and performance of offspring born to dams supplemented with fat during late gestation was the long-term impact reported by Santos et al. [69]. These researchers [69] reported an improvement in milk yield during the first lactation of heifers born from dams supplemented with lipids on late gestation dams. As stated by Santos et al. [69], heifers from dams fed fat prepartum produced 1400 kg more 305-d mature equivalent milk yield in the first lactation than those fed diets with no supplemental fat. Furthermore, although calves born from dams fed SFA gained more BW than those born from dams fed PUFA, heifers born from PUFA cows had numerically greater (517 kg) 305-d mature equivalent milk production than heifers born from SFA dams.

Regarding carcass traits, n-3 + n-6 PUFA supplementation of late gestating beef cows has also shown an improvement in carcass quality (greater hot carcass weight; HCW), greater longissimus dorsi muscle (LM) area, and greater marbling score of their offspring [70] when compared to calves born to dams supplemented with SFA + MUFA (Table 1). Moreover, Marques et al. [70] stated that calves born from n-3 + n-6 PUFA supplemented cows also tended to have a greater percent of carcasses graded as Choice, although most of calves from SFA + MUFA supplemented cows also graded Choice. Brandão et al. [76] also reported a treatment effect for carcass LM area, which was greater in calves from n-6 cows compared with those born to SFA cows across sexes. These results are also indicative of the fetal programming effects from supplementing PUFA to late gestating beef cows. A treatment by sex interaction was also reported for carcass traits in two ruminant models [71,76]. In beef cattle, a treatment by sex interaction for feed yard HCW was observed [76], where steers from n-6 cows showed a heavier HCW compared with those born to SFA supplemented cows and did not differ between heifers. In sheep [71], wethers born from PUFA ewes had the heaviest HCW when compared to the rest of the lambs in the other two treatments (MUFA and no supplemented). The reason for the treatment by sex interaction in HCW reported in these studies [10,76] remains unclear, since males and females in the different treatment groups were equal.

The mechanisms behind the changes observed in growth and carcass traits of the offspring have not been completely elucidated. Nevertheless, some of these studies showed an association between the observed increase in offspring BW with an increase in feed intake [65,69], with alterations in mRNA expression [6,7,10], DNA methylation [10], changes in markers of energy metabolism (in the glucose-insulin system; [8,71], and molecules involved in the inflammatory response (like haptoglobin [70], IgG [69], and resolvin D1 (RvD1; [71]).

#### 3.3.1. Effect on mRNA Expression and DNA Methylation of the Offspring

Maternal supplementation with FA in late gestation reported changes in global DNA methylation [10], and mRNA expression of genes associated with the immune response, myogenesis, and metabolism in different tissues [7,10,76,82] of the offspring during the prenatal and postnatal periods (Table 1). These responses have been associated with changes in growth performance of the offspring. In a study conducted in sheep, Rosa-Velazquez et al. [10] reported an increase in fetal hepatic global DNA methylation in fetuses from ewes supplemented with a source of n-3 PUFA during late gestation compared with fetuses from non-supplemented dams. In addition, n-3 PUFA supplementation increased the mRNA relative expression of *arachidonate-5-lipoxygenase-activating-protein* (*ALOX 5AP*) which is an enzyme involved in the last step of formation of an anti-inflammatory lipid mediator (resolvin D1). These changes in hepatic global DNA methylation and mRNA relative expression of *ALOX 5AP* were associated with an increase of fetal liver weight [71]. Supplementation with n-3 PUFA also increases DNA methylation in the small intestine [83]. These changes were associated with an increase in mRNA and protein expression of amino acid transporters in the small intestine [83]. This might be associated with an increase in nutrient absorption, but, currently, there are no studies that evaluate the effect of FA and developmental program on nutrient digestibility in ruminants.

In a study in beef cattle [76], calves born to n-6 supplemented cows also had greater LM mRNA expression of myogenic differentiation 1 (MyoD) and myogenin genes in the LM at birth. This observed increase in MyoD mRNA expression could indicate a greater proliferation of myocytes in the LM of the offspring at birth, which terminally develop into muscle fibers upon myogenin expression [84,85], since MyoD is a regulatory factor expressed by myocytes that regulate postnatal muscle growth through differentiation and fusion with existing muscle fibers [86,87].

In a study in dairy cattle [82] where cows were fed a diet containing either a source of n-6 or a source of SFA during the last eight weeks of gestation, and hepatic biopsies were taken on the offspring during the preweaning period, dam FA supplementation, and type of fatty acid supplementation modified mRNA expression of genes involved in biological functions and pathways associated downregulation of inflammatory responses. They concluded [82] that the lower expression of pro-inflammatory genes in calves born from dams supplemented with lipids could indicate either a better ability by these calves to minimize an excessive proinflammatory state, thus preventing a negative effect of chronic inflammation on calf growth performance and productivity.

In sheep, the concentration of mRNA hormone receptors, neuropeptides, and their receptors were measured in hypothalamus biopsies collected during the finishing period from female lambs born to late gestating ewes supplemented with either a source of n-3 PUFA or SFA and MUFA [7]. Lambs born from n-3 PUFA ewes had a lower concentration of MCR4 mRNA than the ones born from SFA + MUFA supplemented ewes. As noted by Carranza-Martin et al. [7], previous studies have reported that a mutation in MCR4 increases body weight [88] due to its role in regulating appetite [89]. Carranza Martin et al. [7] reported that the increase in the melanocortin receptor was not associated with differences in feed intake, but that the mRNA concentration of this melanocortin receptor was associated with a decrease in body weight. These investigators [7] stated that they could not prove that there was a cause/effect on these two variables, but their results suggest that changes in the mRNA expression on genes involved in appetite regulation could affect offspring growth performance during the finishing period.

#### 3.3.2. Effect in Offspring’s Energy Metabolism

The effect of maternal FA supplementation during late gestation has reported contradictory results regarding its effect on offspring energy metabolism at birth, through weaning, and during the finishing period. Nonetheless, changes in offspring energy metabolism born to dams supplemented with a source of FA during late gestation have been associated with changes in offspring growth. Increasing doses (concentrations of 0, 1, or 2% of dry matter intake during the last 50 d of gestation) of a source of n-3 PUFA in the diet of ewes in the last third of gestation reported an increase in plasma glucose concentration and a tendency to decrease ghrelin concentrations in the offspring during the finishing period [8]. The findings reported by Nickles et al. [8] contradicted other studies [5,10,65]; therefore, to understand this difference, we conducted an experiment using a glucose tolerance test (GTT) during the finishing period [71]. In this study [71], plasma concentration of glucose, insulin, and ghrelin were measured. Different from what was reported by Nickles et al. [8], dam FA supplementation did not have any effect on plasma glucose or ghrelin concentration during the GTT. On the other hand, there was an FA by time by sex interaction for plasma insulin concentration [71] (Table 1). Wethers plasma insulin concentration tended to increase as FA unsaturation degree increased during the GTT, while the plasma insulin concentration decreased with increasing FA unsaturation for female lambs. Ewe lambs born from dams supplemented with MUFA during the last 50 days of gestation tended to have a greater insulin concentration than wethers born from MUFA supplemented ewes [71]. As aforementioned, the results in plasma insulin concentration reported by Rosa-Velazquez et al. [71] were positively associated with greater finishing body weight as reported by Nickles et al. [8]. Interestingly, these investigators reported a dam FA supplementation and offspring sex interaction in sheep, which was not been reported in previous studies; dam FA supplementation during late gestation affected the glucose-insulin system in a sex-dependent manner, suggesting that males born from PUFA supplemented ewes are less sensitive to insulin than females born to ewes fed the same FA during gestation, and this lack in insulin response was similar in female lambs born from MUFA supplemented ewes [71].

#### 3.3.3. Effects in Offspring’s Immune Response and Inflammatory Markers

It has been reported that supplementation during late gestation with n-3 PUFA [90] can modulate factors involved in the immune response having effects on the health as subsequent growth of the offspring. In ruminants, changes in molecules involved in the immune response and inflammatory markers of the offspring have been associated with an improvement in growth after maternal FA supplementation during late gestation [65,69,70,71,76,81]. In the study by Santos et al. [69], where two FA supplementations were fed [PUFA (linoleic, n-6, and alpha-linolenic, n-3) or SFA, it was reported that FA supplementation during late gestation to dairy cows can improve the transfer of passive immunity to the progeny (Table 1). In this study [69], the IgG content of colostrum did not differ between treatments; however, total serum IgG and anti-ovalbumin concentrations after calves consumed colostrum were greater in animals born to cows that consumed FA. In the same way, adding fat to the prepartum diet improved the IgG absorption efficiency from 23.3 to 27.9% regardless of the type of fat fed during late gestation [69]. Two recent studies [76,81] where different sources of PUFA were supplemented during late gestation in cattle reported an improvement in the immunity and health indices of the offspring born to FA supplemented dams (Table 1). Plasma concentrations of IgG and apparent efficiency of IgG absorption were improved in calves born from dams fed fat (n-6 or n-3) after colostrum consumption compared with those born to non-supplemented cows [81]. These findings support those reported by Garcia et al. [65] who reported that calves born from dams fed fat (saturated or rich in n-6 FA) tended to have greater serum concentrations of total IgG and better apparent efficiency of IgG absorption after colostrum feeding than calves born from dams fed non-supplemental fat. Jolazadeh et al. [81] also found that supplementation with fat during late gestation reduced rectal temperature during a pre-weaning period when compared with those born to non-supplemented cows. Similarly, Brandão et al. [76] reported greater concentrations of IgG in calf plasma 24 h after birth in calves born to dams supplemented with a source of n-6 compared to calves born from cows supplemented with SFA. Moreover, these researchers [76] found that the incidence of calves diagnosed with bovine respiratory disease (BRD) that required a second antimicrobial treatment was less in calves from n-6 supplemented cows, which resulted in a reduced need for treatments to regain health compared with the calves born to cows supplemented with SFA. According to their findings, Brandão et al. [76] indicated that calves born to n-6 supplemented cows displaced improved immunocompetence in feed yard entry when BRD incidence is typically elevated [91]. Thus, the observed improvement in calf immunity could be attributed to their greater plasma IgG concentrations 24 after birth, which could positively impact calf immunity later in life [92]. Brandão et al. [76] stated that calves from n-6 cows were more likely to have a greater ability to absorb colostrum IgG, as PUFA incorporated into intestinal cells upregulates IgG receptors (e.g., neonatal Fc receptor) responsible for IgG absorption in neonates [93,94]. Another recent study [77] also reported greater serum IgG concentrations 24 h after birth in calves born from cows supplemented with n-6 during late gestation and suggested that increased passive transfer of IgG due to maternal n-6 supplementation could help to mitigate subsequent calf morbidity and mortality. As stated previously, no studies are evaluating how gut physiology changes in FA programed ruminants.

Fisher-Heffernan et al. [90] observed that ewe supplementation during late gestation with a source of n-3 PUFA could protect against adverse fetal programming that may occur during maternal infection and thus may reduce the risk of atopic disease later in life. Ewes were fed a diet supplemented either with fishmeal (source of n-3 PUFA) or soybean meal (source of n-6 PUFA) from day 100 of gestation through lactation. On day 135 of gestation, half of the ewes from each dietary group were challenged with either 1.2 μg/kg Escherichia coli lipopolysaccharide (LPS) endotoxin to simulate a bacterial infection, or saline as the control. Lambs’ dermal immune response was assessed by cutaneous hypersensitivity testing with ovalbumin (OVA) and candida albicans (CAA) 21 days after sensitization at 4.5 months of age [90]. It was found that ewe n-3 PUFA supplementation during late pregnancy protected the offspring from maternal endotoxin challenge and decreased the dermal immune response and antibody-specific response to novel antigens. It is important to point out that this was the first study to investigate whether n-3 supplementation during gestation and lactation can protect the offspring from programming following simulated maternal infection during late gestation. The Fisher-Heffernan et al. [90] study is different from others as supplementation with n-3 PUFA occurred solely through the dam and the offspring n-3 PUFA concentrations were not different across treatments at the time of challenge with OVA and CAA.

In a recent study conducted in our lab [71], wethers born from n-3 supplemented dams during late gestation had a greater RvD1 concentration than females from the same treatment; and ewe lambs born from MUFA supplemented dams showed a greater RvD1 plasma concentration than males from the same treatment. In addition, a study conducted by Rosa-Velazquez et al. [10] reported a greater mRNA expression of *arachidonate-5-lipoxygenase-activating-protein*, an enzyme that drives the final formation step of RvD1 [95], in fetal livers from dams that were supplemented with n-3 PUFA during the last third of gestation (Table 1). To our knowledge, the study conducted by Rosa-Velazquez et al. [71] was the first to evaluate the effect of dam’s FA supplementation on offspring plasma RvD1 concentration. Their findings [71] suggested that offspring plasma RvD1 concentration at birth could be affected by a source of dietary FA supplemented during gestation and the sex of the newborn. Moreover, the changes observed in RvD1 concentration at birth in this study were positively associated with changes in offspring body weight during the finishing period. Hence, the possible modulatory effect of maternal PUFA supplementation during late gestation on newborns’ inflammatory response could have had a positive effect on the lamb’s growth during the finishing period. Additionally, a study conducted in beef cattle reported that maternal supplementation with PUFA (n-3 + n-6) during late gestation affected the offspring’s plasma haptoglobin concentration, a protein involved in the acute-phase reaction [70]. The findings reported by several authors suggest that maternal supplementation with PUFA during late gestation might modulate the offspring’s inflammatory response, which could result in lasting effects in the offspring growth performance and health. We understand that the regulation of inflammation is a complex mechanism, and the measurements of markers do not fully explain the entire process. However, it is possible that FA supplementation during gestation can program the fetus to have a more specific and short response to certain injuries, which allowed them to improve growth.

## 4. Effect of FA Supplementation on Early Life for the Developmental Program

We can consider that the developmental program does not finish at birth but continues in the early life of the animals [53]. Fatty acid supplementation during early life in ruminants has reported beneficial effects in the individual’s metabolism, growth, and health performance (Table 1). Santos et al. [69] reported that increased intake of linoleic acid from approximately 6.2 to 13.2 g/d on average over a 60-day period preweaning by feeding a high linoleic acid milk replacer increased body weight gain by 3 kg over the 60-day period. Moreover, since feed intake was not changed, the conversion of feed to gain was improved by 8%. This observed enhanced performance was accompanied by an increased plasma concentration of glucose, and in the proportion of phagocytosis by blood neutrophils and a greater synthesis of cytokines by blood mononuclear cells. Thus, feeding more linoleic acid in the milk replacer influenced growth, metabolism, and immune cell function. In another study [81] conducted on dairy cattle, calves fed a calf starter rich in unsaturated FA had a greater average daily gain, skeletal growth, feed efficiency, and weaning weight compared with non-supplemented calves. Furthermore, calves fed unsaturated FA had lower rectal temperature during the pre- and post-weaning periods and fewer days with diarrhea compared with non-supplemented calves. However, in beef cattle, Schubach et al. [96] supplemented the calves preweaning with different sources of FA and did not observe differences in growth. Capper et al. [73] supplemented ewes during late gestation and lactation, and they observed an increase in n-3 PUFA in the dam milk, but it was associated with a decrease in growth in the lambs (Table 1). Part of the differences in animal growth could be associated with the time when the FA were supplemented or a different animal model. None of the papers on early life FA programming [73,81,96] report any physiological mechanism for these differences; however, more studies should be conducted to elucidate physiological differences.

## 5. Conclusions

The results presented in this review, and reported by several authors, are indicative of the programming effects on pre- and postnatal offspring growth and health indices resulting from FA supplementation to ruminants, although the mechanism underlying these effects, including the specific role of n-3 and n-6 PUFA in offspring growth performance and physiology, also requires investigation. Nevertheless, the changes in growth performance observed in the offspring born FA supplemented dams could be related to changes in the glucose-insulin system and molecules involved in the inflammatory response (protein- and lipid-derived mediators). These outcomes are novel and indicate FA supplementation during late gestation might be a feasible alternative to optimize offspring health and subsequent productivity. Some areas need further investigation, such as pre- or post-gestational development, or how and why different fatty acids have different effects depending on the stage of gestation.

## Figures and Tables

**Figure 1 animals-11-00762-f001:**
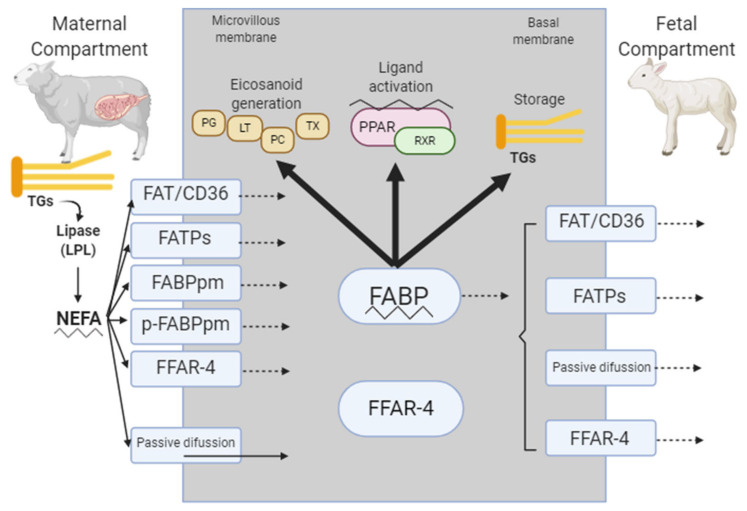
Placental fatty acid transport mechanisms and potential metabolic fates within the placental tissue (modified from [54,62] and add the data from Roque-Jimenez et al. [9]) TGs, triglycerides; LPL, lipoprotein lipase; EL, epithelial lipase; NEFA, non-esterified fatty acid; FAT/CD36, fatty acid translocase; FATPs, fatty acid transport proteins; FABP, fatty acid binding proteins (occlude placental (p-) FABP, FFAR-4, free fatty acid receptor 4; PG, prostaglandin; LT, leukotriene; PC, prostacyclin; TX, thromboxane; PPAR, peroxisome proliferator-activated receptor; RXR, retinoic acid X receptor.

**Table 1 animals-11-00762-t001:** Effect of dam FA supplementation on offspring growth and performance.

Reference	Specie—Animal Model	Developmental Period	FA Source (Treatment Comparation)	Offspring Performance
Oviedo-Ojeda et al. [12]	Sheep	First third of gestation	Dam supplementation (DS): 1.6% of Ca salts enriched with SFA + MUFA or EPA+DHA.	Lambs born from DS with SFA + MUFA have a greater DMI. Lambs born from DS with SFA + MUFA and fed EPA+DHA during the finishing period had a greater growth rate.
Carranza-Martin et al. [7]	Sheep	Last third of gestation	Dam supplementation (DS): a diet containing 0.39% (dry matter basis) Ca salts enriched with SFA + MUFA or EPA+DHA.	Lambs born from EPA+DHA dams were heavier at the start of the finishing period.
Nickles et al. [8]	Sheep	Last third of gestation.	Dam supplementation (DS): diets with Ca salts enriched with EPA and DHA at concentrations of 0, 1 or 2% of DMI.	Lambs born from DS with EPA and DHA increasing DMI, ADG and BW. Lambs from ewes fed with EPA and DHA had increased plasma glucose concentration and tended to decrease plasma ghrelin concentration during the finishing period.
Marques et al. [70]	Beef Cattle	Late gestation	Dam supplementation: (1) 190 g/cow daily of Ca salts of PUFA (EPA+DHA) or (2) 190 g/cow daily of Ca salts of SFA + MUFA based on palmitic and oleic acids.	During both the growing and finishing period, ADG was greater in calves from PUFA-supplemented cows. Upon slaughter, hot carcass weight and marbling were also greater in calves from PUFA-supplemented cows.
Garcia et al. [74]	Dairy Cattle	Late Gestation (8 weeks before calculated parturition date)	Dam supplementation: (1) no fat supplement, (2) 1.7% SFA supplement, or (3) 2.0% of PUFA (EPA+DHA).	Prepartum supplementation of SFA tended to improve intake of grain from 31 to 60 d of life and improved ADG of calves.
Rosa-Velazquez et al. [83]	Sheep	Last 50 days of gestation	Dam supplementation: (1) no FA (NF); (2) a source of SFA +MUFA (1.01 % of Ca salts); or (3) a source of PUFA (1.01 % of Ca salts containing EPA and DHA).	Females’ offspring from MUFA were heavier than MUFA males, while PUFA offspring males were heavier than PUFA at the finishing stage. Plasma insulin concentration of males increased as FA unsaturation degree increased during the GTT; the opposite happened with female lambs.
Bellows et al. [75]	Beef cattle	Late gestation (last 68.2 ± 5.5 d before calving)	Dam diets: control or added sunflower seeds.	Calf birth BW from sunflower seeds dams tended to be heavier.
Ricks et al. [77]	Beef cattle	Late gestation	Dam diets: (1) No Fat or (2) 200 g Essential FA (EFA) (Essentiom, Church and Dwight Co., Princeton, NJ).	Calf BW up to weaning was increased in calves from second- and third-parity EFA dams.
Brandão et al. [76]	Beef cattle	Late gestation	Dam supplementation: (1) SFA or (2) n-6 PUFA (Ca salts of soybean oil).	Average daily gain and final BW in the feed yard were greater in steers from PUFA.
Banta et al. [79]	Beef cattle	Late gestation (for an average of 83 d during mid to late gestation)	Dam supplementation: (1) soybean hull-based supplement; (2) linoleic sunflower seed, and (3) mid-oleic sunflower seed.	No differences were detected in calf birth or weaning BW.
Banta et al. [80]	Beef cattle	During mid to late gestation	Dam supplementation: (1) soybean meal/feeding; (2) soybean hull-based supplement; and (3) whole sunflower seeds high in linoleic acid.	No difference among treatments was detected for calf birth weight, or calf weaning weight. Supplements fed to dams during gestation did not influence feedlot performance or carcass characteristics.

## Data Availability

No new data were created or analyzed in this study. Data sharing is not applicable to this article.

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
