# Peer review of "Role of Long Chain Fatty Acids in Developmental Programming in Ruminants"

_animals, 2021, doi:10.3390/ani11030762_

Round 1
Reviewer 1 Report
The article entitled “Role of long-chain fatty acids in developmental programming in ruminants” needs to be revised properly and corrected before publication. The following points need to be corrected and improved:
- In the “Introduction” section: Please mention the species of the experimental model such as cattle/sheep/goat/human according to the reference articles. Please see the comments in the attached file.
- Please check the reference according to the statement.
- English grammar should be corrected properly. A huge mistake in the use of articles (a/an/the) in the manuscript. Please edit the manuscript for the English language.
- Please revise the manuscript carefully according to the suggestion (see the attached file).
- Please revise and edit the supplementary page for the English.

Author Response
Dear Reviewer. Thank you very much for your time reviewing our manuscript. Please find attach the response to your comments.
Regards
Ale Relling

Reviewer 2 Report
Comments to the Authors of manuscript number: animals-1105439 entitled “Role of long chain fatty acids in developmental programming in ruminants”.
Authors have presented the knowledge concerning to PUFA in fetal program, which determined not only prenatal development but determine the growth, productivity and all aspects of postnatal life.
Congratulations to the authors of an extensive review, which collected a lot of information on a very difficult topic. It is a pleasure to read.
- What is the difference between fetal program and prenatal programming firstly called imprinting?
Maybe it worth to mention about all these terms.
- L 63 Just this imbalance results in the alteration of intrauterine circumstances during prenatal life, which trigger structural alteration and functional changes on the level of cells, tissues, organs and even systems during prenatal development. Thus in order to adapt to changes circumstance as an effect of prenatal programming fetal metabolism is changed. It increases the chance to keep fetus alive until delivery, but on the other hand all these events can predispose to some health problems later in life. Of course, there are some aspects positive and negative of prenatal or fetal program.
It should be explained.
- L-79, L 83 because the Animals is the Journal concerning only animals, please avoid information about humans. If the mention is needed it is worth rather to write - vertebrae.
- L 112- humans should be omitted. The paper concerns to ruminant. The comparison both species is difficult due to the presence of the rumen, reticulum, and omasum. Thus, what is their role in the PUFA intake in ruminants? What is happened with PUFA in the rumen, which is abundant in various species of microorganisms?
- L 112 – avoid human
- What is relationship between PUFA and particular species of microorganisms, the likely pathway by which nutrition can influence the ruminant development.
- L 211 – I am not sure if the mention of human is proper
Author Response
Dear Reviewer, thanks for you comments, all of them were addressed and in the attached letter we include our responses to your comments.
Regards
Ale Relling

Round 2
Reviewer 1 Report
Please delete the verb "has" on line nr 589 in your corrected manuscript.
Author Response
We thanks the meticulous editorial comments of this reviewer. The word has in line 589 was removed. We also removed the highlighted areas of the main document and the Supplementary table.